# Speaking out of turn: How video conferencing reduces vocal synchrony and collective intelligence

**Maria Tomprou** [1]*, **Young Ji Kim**[2], **Prerna Chikersal**[3], **Anita Williams Woolley**[1], **Laura A. Dabbish**[3]

**1** Tepper School of Business, Carnegie Mellon University, Pittsburgh, Pennsylvania, United States of America, **2** Department of Communication, University of California, Santa Barbara, Santa Barbara, California, United States of America, **3** Human-Computer Interaction Institute, Carnegie Mellon University, Pittsburgh, Pennsylvania, United States of America

* mtomprou@cmu.edu

**Data Availability Statement:** The data of the study are publicly available at https://osf.io/tnv93/.

**Funding:** This material is based upon work supported by the National Science Foundation under grant numbers CNS-1205539 (url: https://

## Abstract

Collective intelligence (CI) is the ability of a group to solve a wide range of problems. Synchrony in nonverbal cues is critically important to the development of CI; however, extant findings are mostly based on studies conducted face-to-face. Given how much collaboration takes place via the internet, does nonverbal synchrony still matter and can it be achieved when collaborators are physically separated? Here, we hypothesize and test the effect of nonverbal synchrony on CI that develops through visual and audio cues in physically-separated teammates. We show that, contrary to popular belief, the presence of visual cues surprisingly has no effect on CI; furthermore, teams *without* visual cues are more successful in synchronizing their vocal cues and speaking turns, and when they do so, they have higher CI. Our findings show that nonverbal synchrony is important in distributed collaboration and call into question the necessity of video support.

## Introduction

In order to survive, members of social species need to find ways to coordinate and collaborate with each other [1]. Over a number of decades, scientists have come to study the collaboration ability of collectives within a framework of collective intelligence, exploring the mechanisms that enable groups to effectively collaborate to accomplish a wide variety of functions [2–6].

Recent research demonstrates that, like other species, human groups exhibit "collective intelligence" (CI), defined as a group's ability to solve a wide range of problems [2, 3]. As humans are a more cerebral species, researchers have thought that their group performance depends largely on verbal communication and a high investment of time in interpersonal relationships that foster the development of trust and attachment [7, 8]. However, more recent research on collective intelligence in human groups illustrates that it forms rather quickly [2], is partially dependent on members' ability to pick up on subtle, nonverbal cues [9–11], and is strongly associated with teams' ability to engage in tacit coordination, or coordination without

www.nsf.gov/awardsearch/showAward?AWD_ID=1205539&HistoricalAwards=false) Author who received the award: L.D., OAC-1322278 (url:https://nsf.gov/awardsearch/showAward?AWD_ID=1322278) (Author who received the award A.W.), and OAC-1322254 (url:.https://nsf.gov/awardsearch/showAward?AWD_ID=1322254) (Author who received the award A.W.). The funders had no role in study design, data collection and analysis, decision to publish, or preparation of the manuscript.

**Competing interests:** The authors have declared that no competing interests exist.

verbal communication [12]. This suggests that there is likely a so-called deep structure to CI in human groups, with nonverbal and physiological underpinnings [12, 13], just as is the case in other social species [14, 15].

Existing research suggests that nonverbal cues, and their synchronization, play an important role in human collaboration and CI [10]. Nonverbal cues are those that encompass all the messages other than words that people exchange in interactive contexts. Researchers consider nonverbal cues more reliable than verbal cues in conveying emotion and relational messages [16] and find that nonverbal cues are important for regulating the communication pace and flow between interacting partners [17, 18]. The literature on interpersonal coordination explores many forms of synchrony [19, 20], but the common view is that synchrony is achieved when two or more nonverbal cues or behaviors are aligned [21, 22]. Social psychology researchers traditionally study synchrony in terms of body movements, such as leg movements [23], body posture sway [24, 25], finger tapping [26] and dancing [27]. These forms of synchrony contribute to interpersonal liking, cohesion, and coordination in relatively simple tasks [28, 29]. Synchrony in facial muscle activity [30] and prosodic cues such as vocal pitch and voice quality [31–33] are of particular importance for the coordination of interacting group members, as these facilitate both communication and interpersonal closeness. For example, synchrony in facial cues has been consistently found to indicate partners' liking for each other and cohesion [30].

While humans in general tend to synchronize with others, interaction partners also vary in the level of synchrony they achieve. The level of synchrony in a group can be influenced by the qualities of existing relationships [34] but can also be influenced by the characteristics of individual team members; for instance, individuals who are more prosocial [35] and more attentive to social cues [10, 36] are more likely to achieve synchrony and cooperation with interaction partners. And, consistent with the link between synchrony and cooperation, recent studies demonstrate that greater synchrony in teams is associated with better performance [37, 38].

Among the elements that nonverbal cues coordinate is spoken communication, particularly conversational speaking turns, wherein partners regulate nonverbal cues to signal their intention to maintain or yield turns [39]. Conversational turn-taking has fairly primitive origins, being observed in other species and emerging in infants prior to linguistic competence, and is evident in different spoken languages around the world [40]. The equality with which interaction partners speak varies, however, and those who do have more speaking equality consistently exhibit higher collective intelligence [2, 11]. The negative effect of speaking inequality on collective intelligence has been demonstrated both in face-to-face and online interactions [11].

The majority of existing studies on synchrony were conducted in face-to-face environments [20, 30, 41] and focused on the relationship between synchrony and cohesion. We have a limited understanding of how synchrony relates to collective intelligence, particularly when group members are not collocated and collaborate on an *ad hoc* basis -a form of modern organization that has become increasingly common [42, 43]. Given the exponential growth in the use of technology to mediate human relationships [44, 45], an important question is whether synchrony in common, nonverbal communication cues in face-to-face interaction, such as facial expression and tone of voice, still plays a role in human problem-solving and collaboration in mediated contexts, and how the role of different cues changes based on the communication medium used.

Researchers and managers alike assume that the closer a technology-mediated interaction is to face-to-face interaction–by including the full range of nonverbal cues (e.g., visual, audio, physical environment)–the better it will be at fostering high quality collaboration [46–48]. The

idea that having more cues available helps collaborators bridge distance is strongly represented in both the management literature [49, 50] and lay theory [51]. However, some empirical research suggests that visual cue availability may not always be superior to audio cues alone. In the absence of visual cues, communicators can effectively compensate, seek social information, and develop relationships in technology-mediated environments [52–55]. Indeed, in some cases, task-performing groups find their partners more satisfactory and trustworthy in audio-only settings than in audiovisual settings [56, 57], suggesting that visual cues may serve as distractors in some conditions.

## Purpose of the study and hypotheses

The primary goal of this research is to understand whether physically distributed collaborators develop nonverbal synchrony, and how variation in audio-visual cue availability during collaboration affects nonverbal synchrony and collective intelligence. Specifically, we test whether nonverbal synchrony–an implicit signal of coordination–is a mechanism regulating the effect of communication technologies on collective intelligence. Previous research defines nonverbal synchrony as any type of synchronous movement and vocalization that involves the matching of actions in time with others [23]. This study focuses on two types of nonverbal synchrony that are particularly relevant to the quality of communication and are available through virtual collaboration and interaction–namely, facial expression and prosodic synchrony. We hypothesize that in environments where people have access to both visual and audio cues, collective intelligence will develop through facial expression synchrony as a coordination mechanism. When visual cues are absent, however, we anticipate that interacting partners will reach higher levels of collective intelligence through prosodic synchrony. It will also be interesting to see if facial expression synchrony develops and affects collective intelligence even in the absence of visual cues; if this occurs, it would suggest that this type of synchrony forms, at least in part, based on similarity in partners' internal reactions to shared experiences, versus simply as reactions to partner's facial expressions. If facial expression synchrony is important for CI only when partners see each other, it would suggest that the expressions play a predominantly social communication role under those conditions, and the joint attention of partners to these signals is an indicator of the quality of their communication. To explore these predictions, we conducted an experiment where we utilized two different conditions of distributed collaboration, one with no video access to collaboration partners (Condition 1) and one with video access (Condition 2) to disentangle how the types of cues available affect the type of synchrony that forms and its implications for collective intelligence.

## Method

### Participant recruitment and data collection

Our sample included 198 individuals (99 dyads; 49 in Condition 1 and 50 in Condition 2). We recruited 292 individuals from a research participation pool of a northeastern university in the United States and randomly assigned into 146 dyads (59 in condition 1 and 87 in condition 2). Due to technical problems with audio recording, ten dyads had missing audio data in Condition 1 and 37 dyads in Condition 2 resulting in 62% valid responses. To test for possible bias introduced by missing data, we conducted independent sample t-tests to assess any differences in demographics between the dyads retained and those we excluded due to technical difficulties; no differences were detected (see S1 Appendix). All signed an informed consent form. The average age in the sample was 24.82 years old (SD = 7.18 years); Ninety-six participants (48.7%) were female. The ethnic composition of our sample was racially diverse: 6.6% from different races, 50% Asian or Pacific, 33% White or Caucasian, 7% Black or African American,

**Facial Expressions**

**Fig 1. This flowchart illustrates the methodology used to transform the raw data of each participant into individual signals or measures from which synchrony and spoken communication features are calculated.**

2.5% Latin or Hispanic. Carnegie Mellon University's Institutional Review Board approved all materials and procedures in our study. The participant in Fig 1 has provided a written informed consent to publish their case details.

The procedure was the same in both conditions, except that in Condition 1 there was no camera and participants could only hear each other through an audio connection. In Condition 2, participants could also see each other through a video connection. Both conditions had approximately equal numbers of dyads in terms of gender composition (i.e., no female, one female, only-female dyads). Each session lasted about 30 minutes. Members of each dyad were seated in two separate rooms. After participants completed the pre-test survey independently, they initiated a conference call with their partner. Participants logged onto the Platform for Online Group Studies (POGS: pogs.mit.edu), a web browser-based platform supporting synchronous multiplayer interaction, to complete the Test of Collective Intelligence (TCI) with their partner [2, 11]. The TCI contained six tasks ranging from 2 to 6 minutes each, and instructions were displayed before each task for 15 seconds to 1.5 minutes. At the end of the test, participants were instructed to sign off the conference call. Participants were then compensated and debriefed. The publication has created a laboratory protocol with DOI.

## Measures

**Collective intelligence.** Collective intelligence was measured using the Test of Collective Intelligence (TCI) completed by dyads working together. The TCI is an online version of the collective intelligence battery of tests used by [2], which contains a wide range of group tasks [11, 58]. The TCI was adapted into an online tool to allow researchers to administer the test in a standardized way, even when participants are not collocated. Participants completed six tasks representing a variety of group processes (e.g., generating, deciding, executing, remembering) in a sequential order (see study's protocol). To obtain collective intelligence scores for all dyads, we first scored each of the six tasks and then standardized the raw task scores. We then computed an unweighted mean of the six standardized scores, a method adapted from

prior research on collective intelligence [58]. Cronbach's alpha for the reliability of the TCI scores was .81.

**Facial expressions.** We used OpenFace [59] to automatically detect facial movements in each frame, based on the Facial Action Coding System (FACS). We categorized these facial movements as positive (AU12 i.e., lip corner puller with and without AU6 i.e., cheek raiser), negative (AU15 lip i.e., corner depressor and AU1 i.e., inner brow raiser and/or AU4 i.e., brow lowerer) or other expressions (i.e., everything else in low occurrence that may be random). Facial expression synchrony of the dyad is a variable encoding the synchrony between the coded facial expression signals of the partners.

**Prosodic features.** Prosodic characteristics of speech contribute to linguistic functions such as intonation, tone, stress, and rhythm. We used OpenSMILE [60] to extract 16 prosodic features over time from the audio recording of each participant. These features included pitch, loudness, and voice quality, as well as the frame-to-frame differences (deltas) between them. We conducted principal components analysis with varimax rotation and used the first factor extracted, which accounted for 55.87% of the variance in the data. The first factor included four prosodic features: pitch, jitter, shimmer, and harmonics-to-noise ratio. Pitch is the fundamental frequency (or F0); jitter, shimmer, and harmonics-to-noise ratio are the three features that index voice quality [61]. Jitter describes pitch variation in voice, which is perceived as sound roughness. Shimmer describes the fluctuation of loudness in the voice. Harmonics-to-noise ratio captures perceived hoarseness. Previous research has also identified these features as important in predicting quality in social interactions [62]. All features were normalized using z-scores to account for individual differences in range. Speaker diarization was not needed, as the speech of each participant was recorded in separate files.

**Nonverbal synchrony.** Fig 1 illustrates how the raw data of each participant was transformed to derive individual signals or measures. These individual signals or measures were then used to calculate dyadic synchrony in facial expressions and prosodic features, speaking turn inequality, and amount of overall communication. We computed synchrony in facial expressions (coded as positive, negative, and *other* in each frame) and prosodic features between partners for each dyad, using Dynamic Time Warping (DTW). DTW takes two signals and warps them in a nonlinear manner to match them with each other and adjust to different speeds. It then returns the distance between the warped signals. The lower this distance, the higher the synchrony between members of the dyad. Hence, we reversed the signs of the DTW distance measure to facilitate its interpretation as a measure of synchrony. We use DTW instead of other distance metrics such as the Pearson correlation or simple Euclidean distance because DTW is able to match similar behaviors of different duration that occur a few seconds apart, which better captures the responsive, social nature of these expressions (see comparison in Fig 2) For both facial expressions and prosodic features, we calculated synchrony across the six tasks of the TCI.

**Spoken communication.** We computed two features of spoken communication: speaking turn inequality and the amount of overall spoken communication in the dyad. In order to compute features related to the number of speaking turns, we first identified speaking turns in audio recordings of each dyad. All audio frames for which Covarep [63] returned a voicing probability over .80 were considered to contain speech. We extracted turns using the following process [64]. First, only one person can hold a turn at a given time. Each turn passes from person A to person B if person A stops speaking before person B starts. If person B interrupts person A, then the turn only passes from A to B if A stops speaking before B stops. If person A pauses for longer than one second, A's turn ends. When both participants are silent for greater than one second, no one holds the turn. We heuristically chose the threshold of one second, since the pauses between most words in English are less than one second [64]. To measure speaking turn inequality, we computed the absolute difference between the total number of

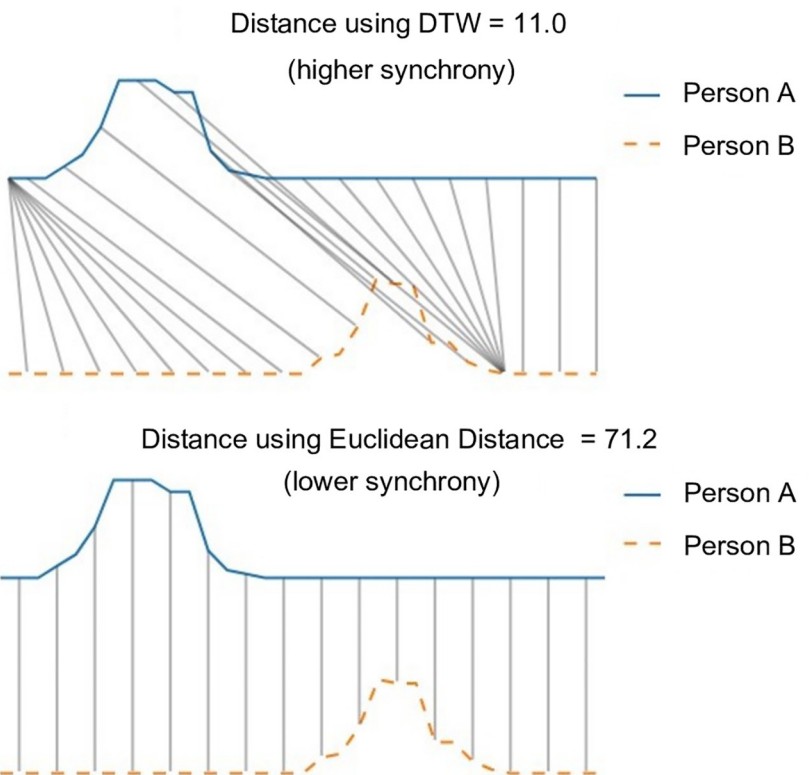

**Fig 2. Dynamic Time Warping (DTW) is a better measure of behavioral synchrony than Euclidean distance because it is able to match similar behaviors of different duration that occur a few seconds apart.**

turns of both partners in the dyad. To measure the amount of overall spoken communication, we summed the total number of samples of speech (i.e., the amount of time each person spoke with voicing probability >.80) of both partners in the dyad.

**Social perceptiveness.** At the beginning of the session, each participant completed the Reading the Mind in the Eyes (RME) test to assess the participant's social perceptiveness [65]. This characteristic gauges individuals' ability to draw inferences about how others think or feel based on subtle nonverbal cues. Previous research has shown that social perceptiveness enhances interpersonal coordination [66] and collective intelligence [2, 11]. The test consists of 36 images of the eye region of individual faces. Participants were asked to choose among possible mental states to describe what the person pictured was feeling or thinking. The options were complex mental states (e.g., guilt) rather than simple emotions (e.g., anger). Individual participants' scores were averaged for each dyad. We controlled for social perceptiveness in our analyses predicting CI, because it is a consistent predictor of collective intelligence in prior work.

**Demographics.** We also collected demographic attributes such as race, age, education, and gender for each participant. As our level of analysis was the dyad, we calculated race similarity, age and education distance, and number of females in the dyad.

## Results

Table 1 provides bi-variate correlations among study variables and descriptive statistics. We first examined whether collective intelligence differs as a function of video availability. An

**Table 1. Correlation matrix for study variables and descriptive statistics.**

|  | 1 | 2 | 3 | 4 | 5 | 6 | 7 | 8 | 9 | 10 | 11 |
|---|---|---|---|---|---|---|---|---|---|---|---|
| 1. Collective intelligence |  |  |  |  |  |  |  |  |  |  |  |
| 2. Facial expression synchrony | .16 |  |  |  |  |  |  |  |  |  |  |
| 3. Prosodic synchrony | .29** | .02 |  |  |  |  |  |  |  |  |  |
| 4. Speaking turn inequality | -.13 | .10 | -.35** |  |  |  |  |  |  |  |  |
| 5. Overall spoken communication | -.24* | -.05 | -.10 | -.11 |  |  |  |  |  |  |  |
| 6. Video condition | -.12 | -.05 | -.28** | .46** | -.16 |  |  |  |  |  |  |
| 7. Social perceptiveness | .33** | .08 | .02 | .03 | .02 | -.04 |  |  |  |  |  |
| 8. Female number | .15 | .04 | .07 | .00 | -.09 | .00 | .20* |  |  |  |  |
| 9. Age distance | -.15 | -.04 | -.04 | .16 | -.06 | .36** | -.18 | -.12 |  |  |  |
| 10. Ethnic similarity | -.02 | -.09 | .00 | -.02 | .08 | .05 | -.22* | -.00 | -.03 |  |  |
| 11. Education distance | -.18 | .10 | -.19 | .05 | -.08 | .05 | -.19 | -.00 | .25* | .09 |  |
| Minimum | -1.64 | -27428 | -3.26 | 0 | 214221 | 0 | 17.5 | 0 | 0 | 0 | 0 |
| Maximum | 1.35 | -1617 | 1.63 | 82 | 16575414 | 1 | 32.5 | 2 | 49 | 4 | 4 |
| Mean | .00 | -7789.28 | 0 | 17.47 | 6765098.17 | - | 26.25 | .98 | 5.64 | .36 | 1.25 |
| SD | .58 | 4206.59 | 1 | 18.44 | 3520702.91 | - | 2.78 | .83 | 7.59 | .48 | 1.14 |

*Note*:

*$p < .05$;

** $p < .01$; N = 99 dyads.

independent samples *t*-test comparing our two experimental conditions (no video vs. video) revealed that there was not a significant difference in the observed level of collective intelligence ($M_{\text{Video}}$ = -.07, $SD_{\text{Video}}$ = .64; $M_{\text{NoVideo}}$ = .08, $SD_{\text{NoVideo}}$ = .53; $t(97)$ = -1.23, $p$ = .22). Further, and surprisingly, the level of synchrony in facial expressions was also not significantly different between the two conditions; dyads with access to video did not synchronize facial expressions more than dyads without access to video ($M_{\text{Video}}$ = -7614.80, $SD_{\text{Video}}$ = 3472.92; $M_{\text{NoVideo}}$ = -7248.58, $SD_{\text{NoVideo}}$ = 3167.11; $t(97)$ = -.55, $p$ = .56). By contrast, the difference in prosodic synchrony between the two conditions was significant; prosodic synchrony was significantly higher in dyads *without* access to video ($M_{\text{Video}}$ = -.32, $SD_{\text{Video}}$ = 1.18; $M_{\text{NoVideo}}$ = .26, $SD_{\text{NoVideo}}$ = .72; $t(97)$ = -2.95, $p$ = .004).

Finally, partners' number of speaking turns were significantly less equally distributed in dyads with video than in dyads with no video (speaking turn inequality $M_{\text{Video}}$ = 26.31, $SD_{\text{Video}}$ = 22.96; $M_{\text{NoVideo}}$ = 9.14, $SD_{\text{NoVideo}}$ = 5.63; $t(97)$ = 5.13, $p$ = .000).

We further examined whether synchrony affects CI differently depending on the availability of video. Though collective intelligence did not differ with access to video, nor did the level of facial expression synchrony achieved, we found that synchrony in facial expressions positively predicted collective intelligence *only* in the video condition (see Fig 3; the unstandardised coefficient for the conditional effect = .0001, $t$ = 2.70, $p$ = .01, bias-corrected bootstrap confidence intervals were between.0000 and.0001, suggesting that when video was available, facial expressions play more of a social role and partners jointly attend to them. Furthermore, social perceptiveness significantly predicted facial expression synchrony in the video condition ($r$ = .31, $p$ = .03), consistent with previous research [10], but not in the no video condition ($r$ = -.17, $p$ = .25).

In addition, in the sample overall we found a main effect of prosodic synchrony on CI; controlling for covariates, prosodic synchrony significantly and positively predicted CI ($b$ = .29, $p$ = .003). We wondered why prosodic synchrony was higher in the no video condition, so we explored other qualities of the dyads' speaking patterns, particularly the distribution in

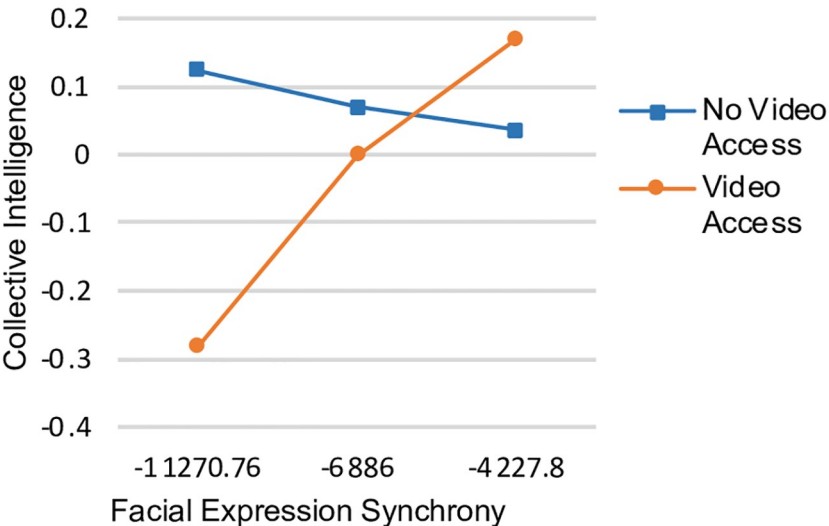

**Fig 3. Interaction effects of facial expression synchrony and video access condition on collective intelligence.**

speaking turns which, as discussed earlier, is an aspect of communication shown to be an important predictor of CI in prior studies [2, 11]. Speaking turn inequality negatively predicted prosodic synchrony, controlling for covariates ($b = -.35$, $p = .001$). Mediation analyses showed that speaking turn inequality mediated the relationship between video condition and prosodic synchrony (effect size = .26, and the bias-corrected bootstrap confidence intervals are between .05 and .44). To test the causal pathway from video access to speaking turn inequality to prosodic synchrony to collective intelligence, we formally tested a serial mediation model. The serial mediation was significant (effect size = .05, and the bias-corrected bootstrap confidence intervals are between -.09 and -.018 (see Fig 4).

That is, video access leads to greater speaking turn inequality and, in turn, decreases the dyad's prosodic synchrony, which then decreases the dyad's collective intelligence (see also Table 2). Note here that an analysis of reverse causality, predicting the speaking turn inequality from prosodic synchrony, was not supported as an alternative explanation.

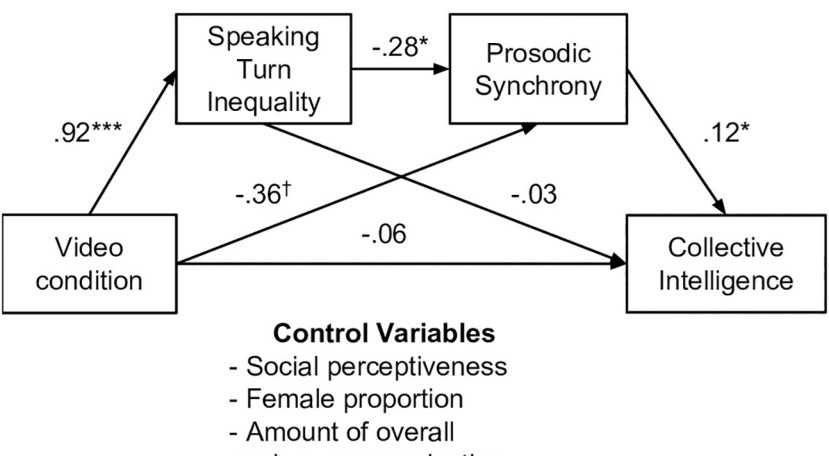

**Fig 4. Serial mediation analysis of the effect of video access on collective intelligence.**

**Table 2. Summary of regression analyses for serial mediation.**

| Dependent Variable: Speaking turn inequality | coefficient | se | t | p | 95% Confidence Intervals | |
|---|---|---|---|---|---|---|
| | | | | | Lower Bound | Upper Bound |
| constant | -.88 | .91 | -.97 | .33 | -2.69 | .92 |
| Social perceptiveness | .02 | .03 | .59 | .55 | -.04 | .08 |
| Female number | -.01 | .11 | -.14 | .88 | -.24 | .20 |
| Overall spoken communication | -.03 | .09 | -.38 | .69 | -.22 | .15 |
| Video condition | .92 | .18 | 4.95 | .00 | .55 | 1.29 |
| $R^2$ = .21, F(4,94) = 6.53, p = .001 | | | | | | |
| Dependent Variable: Prosodic synchrony | coefficient | se | t | p | 95% Confidence Intervals | |
| | | | | | Lower Bound | Upper Bound |
| constant | -.79 | .94 | -.83 | .40 | -2.67 | 1.08 |
| Social perceptiveness | .00 | .03 | .16 | .87 | -.06 | .07 |
| Female number | .06 | .11 | .54 | .58 | -.16 | .29 |
| Overall spoken communication | -.16 | .09 | -1.67 | .09 | -.35 | .03 |
| Video condition | -.36 | .21 | -1.70 | .09 | -.79 | .06 |
| Speaking turn inequality | -.28 | .10 | -2.63 | .00 | -.49 | -.07 |
| $R^2$ = .17, F(5,93) = 3.85, p = .003 | | | | | | |
| Dependent Variable: Collective intelligence | coefficient | se | t | p | 95% Confidence Intervals | |
| | | | | | Lower Bound | Upper Bound |
| constant | -1.90 | .52 | -3.63 | .00 | -2.95 | -8.64 |
| Social perceptiveness | .06 | .01 | 3.51 | .00 | .02 | .10 |
| Female number | .02 | .06 | .45 | .64 | -.09 | .15 |
| Overall spoken communication | -.14 | .05 | -2.58 | .01 | -.25 | -.03 |
| Video condition | -.06 | .12 | -.54 | .58 | -.30 | .17 |
| Speaking turn inequality | -.03 | .06 | -.63 | .52 | -.16 | .08 |
| prosodic synchrony | .12 | .05 | 2.23 | .02 | .01 | .24 |
| $R^2$ = .25, F(6,92) = 5.23, p =.001 | | | | | | |

Note. N = 99 dyads; Video condition coded as 1, No video condition coded as 0.

## Discussion

We explored what role, if any, video access to partners plays in facilitating collaboration when partners are not collocated. Though we found no direct effects of video access on collective intelligence or facial expression synchrony, we did find that in the video condition, facial expression synchrony predicts collective intelligence. This result suggests that when visual cues are available it is important that interaction partners attend to them. Furthermore, when video was available, social perceptiveness predicted facial synchrony, reinforcing the role this individual characteristic plays in heightening attention to available cues. We also found that prosodic synchrony improves collective intelligence in physically separated collaborators whether or not they had access to video. An important precursor to prosodic synchrony is the equality in speaking turns that emerges among collaborators, which enhances prosodic synchrony and, in turn, collective intelligence. Surprisingly, our findings suggest that video access may, in fact, impede the development of prosodic synchrony by creating greater speaking turn inequality, countering some prevailing assumptions about the importance of richer media to facilitate distributed collaboration.

Our findings build on existing research demonstrating that synchrony improves coordination [30, 33] by showing that it also improves cognitive aspects of a group, such as joint

problem-solving and collective intelligence in distributed collaboration. Much of the previous research on synchrony has been conducted in face-to-face settings. We offer evidence that nonverbal synchrony can occur and is important to the level of collective intelligence in distributed collaboration. Furthermore, we demonstrate different pathways through which different types of cues can affect nonverbal synchrony and, in turn, collective intelligence. For example, prosodic synchrony and speaking turn equality seem to be important means for regulating collaboration. Speaking turns are a key communication mechanism operating in social interaction by regulating the pace at which communication proceeds, and is governed by a set of interaction rules such as yielding, requesting, or maintaining turns [18]. These rules are often subtly communicated through nonverbal cues such as eye contact and vocal cues (e.g., back channels), altering volume and rate [18]. However, our findings suggest that visual nonverbal cues may also enable some interacting partners to dominate the conversation. By contrast, we show that when interacting partners have audio cues only, the lack of video does not hinder them from communicating these rules but instead helps them to regulate their conversation more smoothly by engaging in more equal exchange of turns and by establishing improved prosodic synchrony. Previous research has focused largely on synchrony regulated by visual cues, such as studies showing that synchrony in facial expressions improves cohesion in collocated teams [30]. Our study underscores the importance of audio cues, which appear to be compromised by video access.

Our findings offer several avenues for future research on nonverbal synchrony and human collaboration. For instance, how can we enhance prosodic synchrony? Some research has examined the role of interventions to enhance speaking turn equality for decision making effectiveness [67]. Could regulating conversational behavior increase prosodic synchrony? Furthermore, does nonverbal synchrony affect collective intelligence similarly in larger groups? For example, as group size increases, a handful of team members tend to dominate the conversation [68] with implications for spoken communication, nonverbal synchrony, and ultimately collective intelligence. Our results also underscore the importance of using behavioral measures to index the quality of collaboration to augment the dominant focus on self-report measures of attitudes and processes in the social sciences, because collaborators may not always report better collaborations despite exhibiting increased synchrony and collective intelligence [2, 10]. Our study has limitations, which offer opportunities for future research. For example, our findings were observed in newly formed and non-recurring dyads in the laboratory. It remains to be seen whether our findings will generalize to teams that are ongoing or in which there is greater familiarity among members, as in the case of distributed teams in organizations. We encourage future research to test these findings in the field within organizational teams.

Overall, our findings enhance our understanding of the nonverbal cues that people rely on when collaborating with a distant partner via different communication media. As distributed collaboration increases as a form of work (e.g., virtual teams, crowdsourcing), this study suggests that collective intelligence will be a function of subtle cues and available modalities. Extrapolating from our results, one can argue that limited access to video may promote better communication and social interaction during collaborative problem solving, as there are fewer stimuli to distract collaborators. Consequently, we may achieve greater problem solving if new technologies offer fewer distractions and less visual stimuli.

## Supporting information

**S1 Appendix. *t*-test results comparing cases with valid and missing data.**
(PDF)

## Acknowledgments

We thank research assistants Thomas Rasmussen, Brian Hall, and Mikahla Vicino for their help with data collection. We are also grateful to Ella Glickson and Rosalind Chow for providing valuable feedback in earlier versions of this manuscript.

## Author Contributions

**Conceptualization:** Maria Tomprou, Young Ji Kim, Prerna Chikersal, Anita Williams Woolley, Laura A. Dabbish.

**Data curation:** Prerna Chikersal.

**Formal analysis:** Maria Tomprou, Young Ji Kim.

**Funding acquisition:** Anita Williams Woolley, Laura A. Dabbish.

**Investigation:** Maria Tomprou, Prerna Chikersal.

**Methodology:** Maria Tomprou, Prerna Chikersal.

**Project administration:** Laura A. Dabbish.

**Resources:** Anita Williams Woolley, Laura A. Dabbish.

**Software:** Prerna Chikersal, Laura A. Dabbish.

**Supervision:** Anita Williams Woolley, Laura A. Dabbish.

**Writing – original draft:** Maria Tomprou, Young Ji Kim, Prerna Chikersal, Anita Williams Woolley.

**Writing – review & editing:** Maria Tomprou, Young Ji Kim, Prerna Chikersal, Anita Williams Woolley, Laura A. Dabbish.

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
