## [Decision Letter · Decision Letter 0]

24 Nov 2020

PONE-D-20-24495

Visual Cues Disrupt Prosodic Synchrony and Collective Intelligence in Distributed Collaboration

PLOS ONE

Dear Dr. Tomprou,

Thank you for submitting your manuscript to PLOS ONE. After careful consideration, we feel that it has merit but does not fully meet PLOS ONE’s publication criteria as it currently stands. Therefore, we invite you to submit a revised version of the manuscript that addresses the points raised during the review process.

My apologies for the delay getting your manuscript reviewed. I have now received reviews from three referees with various expertise on the topic of your study. All three reviewers are positive about the work, with two recommending acceptance. The second reviewer, however, notes some questions for clarification and offers some constructive suggestions. I agree with the reviewer that these points need to be addressed in order for the manuscript to be suitable for publication. Please pay particular attention to the questions raised about details relating to the methodology of the study.

We look forward to receiving your revised manuscript.

Kind regards,

Marcus Perlman, Ph.D

Academic Editor

PLOS ONE

Journal Requirements:

2. Please ensure that you refer to Figure 4 in your text as, if accepted, production will need this reference to link the reader to the figure.

3.We note that Figure [1] includes an image of a patient / participant in the study. 

4. We noted in your submission details that a portion of your manuscript may have been presented or published elsewhere.

[A pilot of this study was presented at the Computer Supported Cooperative Work and Social Computing Conference in 2017 (please the Related Manuscript). Since that time we have collected additional data, including the addition of Condition 2 in which participants collaborated only via audio. (please see the table of the cover letter

for overlapping variables).]

Reviewers' comments:

Reviewer's Responses to Questions

**Comments to the Author**

1. Is the manuscript technically sound, and do the data support the conclusions?

Reviewer #1: Yes

Reviewer #2: Yes

Reviewer #3: Yes

2. Has the statistical analysis been performed appropriately and rigorously? 

Reviewer #1: Yes

Reviewer #2: Yes

Reviewer #3: Yes

3. Have the authors made all data underlying the findings in their manuscript fully available?

Reviewer #1: Yes

Reviewer #2: Yes

Reviewer #3: Yes

4. Is the manuscript presented in an intelligible fashion and written in standard English?

Reviewer #1: Yes

Reviewer #2: Yes

Reviewer #3: Yes

5. Review Comments to the Author

Reviewer #1: This is a thoughtful, elegant manuscript that meets (and exceeds) the bar for publication--and also offers really interesting, counterintuitive, and practical implications. It is technically sound, and the authors are clearly experts in this field. Excellent submission.

Reviewer #2: #Review of manuscript PONE-D-20-24495 for PLOS ONE

The authors of this work report the results of an experimental study in which they investigate the verbal and nonverbal behaviour and task performance of dyads who carry out several tasks while in different locations (“distributed collaboration”) and connected via audio-only or using concurrent auditory and visual modalities (through live video). They observe that the degree of prosodic synchrony within dyads in the audio-only condition was higher compared to the degree of prosodic synchrony in the audio+video group. No overall differences in collective intelligence measures and facial expression synchrony were observed between the two groups/conditions, but higher prosodic synchrony correlated with higher collective intelligence. The authors argue that the presence of video may thus not only have benefits (as suggested by previous work and anecdotal evidence) but also drawbacks for communication and collaboration, and highlight the societal relevance of these findings in the current digital era. This research is timely, the manuscript is well-written, and I agree that the findings have clear societal relevance. Several questions for clarification keep me from recommending acceptance of the manuscript in its current form.

Minor comments and suggestions

1. The word ‘disrupts’ in the title is quite strong in the absence of a causal effect of the presence of video and in light of the finding that “there was not a significant difference in the observed level of collective intelligence” across the two conditions/groups (line 224).

2. The authors theoretically contrast their set-up with earlier studies that investigated similar issues in ‘face-to-face’ set-ups (see e.g. the abstract, and line 46-48). However one could argue that also in the authors’ video condition, participants are sitting face-to-face – they are just in different locations (see also Figure 1). I would therefore suggest being more careful when using the term ‘face-to-face’ or avoiding it at all.

3. The work by Levinson on turn-taking is relevant to the discussion of turn-taking in the current manuscript and I would suggest referring to it as it has been highly influential. See for instance:

Levinson, S. C. (2016). Turn-taking in human communication–origins and implications for language processing. Trends in Cognitive Sciences, 20(1), 6-14.

4. The authors’ focus on synchronization of facial expressions (line 79) comes a bit as a surprise to the reader at this point in the text. I would suggest adding a paragraph prior to the current line 79 in which earlier work on facial expression synchrony is discussed more explicitly.

5. The authors explain that they used six tasks to measure collective intelligence. More information is needed here in the main text. Which six tasks were used exactly and what was the procedure used in each task? What number (mean unweighted score) corresponds to a high score on those tasks and what corresponds to a relatively low score?

6. In the absence of more explicit information about the tasks that were used, I found it hard to understand why speaking turn inequality (see section 2.2.5) would be a theoretically relevant measure/proxy of a dyad’s spoken communication quality/performance. I would clarify more extensively early on in the manuscript that is has previously been related to collective intelligence, as is now explained in lines 251-252.

7. Very minor stylistic comments: the words co-located and collocated are used interchangeably throughout the manuscript. The final sentence of the abstract seems ungrammatical.

Reviewer #3: The authors investigate the effect of utilizing video versus non-video (audio only) communication on collective intelligence in dyads. Their investigation is nuanced, analyzing the dynamics of mediating and moderating factors in this process.

The design of the experiment is clearly explained, and is sound. The same is true of the analytical strategy.

The results of the study are counterintuitive, but are well-explained by the authors. The idea that the presence of video may actually hinder communication is a fascinating one, and the implications are important. As the authors mention in their discussion of future directions, it will be interesting to see whether these results generalize to larger groups.

In the discussion section, I would like to see a little more discussion/elaboration on the finding displayed in Figure 3. If video (versus audio only) may increase the importance of social perceptiveness (due to its effect on facial expression synchrony), this is an important dynamic for people and organizations to take into consideration.

I enjoyed reading this work - best of luck to all of the authors in the future!

6. PLOS authors have the option to publish the peer review history of their article (what does this mean?). If published, this will include your full peer review and any attached files.

Reviewer #1: No

Reviewer #2: No

Reviewer #3: No

---

## [Author Response · Author response to Decision Letter 0]

23 Jan 2021

PONE-D-20-24495

Visual Cues Disrupt Prosodic Synchrony and Collective Intelligence in Distributed Collaboration

PLOS ONE

Dear Dr. Tomprou,

Thank you for submitting your manuscript to PLOS ONE. After careful consideration, we feel that it has merit but does not fully meet PLOS ONE’s publication criteria as it currently stands. Therefore, we invite you to submit a revised version of the manuscript that addresses the points raised during the review process.

My apologies for the delay getting your manuscript reviewed. I have now received reviews from three referees with various expertise on the topic of your study. All three reviewers are positive about the work, with two recommending acceptance. The second reviewer, however, notes some questions for clarification and offers some constructive suggestions. I agree with the reviewer that these points need to be addressed in order for the manuscript to be suitable for publication. Please pay particular attention to the questions raised about details relating to the methodology of the study.

● A rebuttal letter that responds to each point raised by the academic editor and reviewer(s). You should upload this letter as a separate file labeled 'Response to Reviewers'.

● A marked-up copy of your manuscript that highlights changes made to the original version. You should upload this as a separate file labeled 'Revised Manuscript with Track Changes'.

● An unmarked version of your revised paper without tracked changes. You should upload this as a separate file labeled 'Manuscript'.

We look forward to receiving your revised manuscript.

Kind regards,

Marcus Perlman, Ph.D

Academic Editor

PLOS ONE

Authors’ Response: Thank you for the opportunity to revise and resubmit our paper. We hope we have now carefully and successfully addressed all the comments

Journal Requirements:

Response: After careful editing, we believe we have met all PLOS ONE’s style requirements. 

2. Please ensure that you refer to Figure 4 in your text as, if accepted, production will need this reference to link the reader to the figure.

Response: We apologize for omitting the reference in the actual text. We now have properly referenced all of our Figures. 

3.We note that Figure [1] includes an image of a patient / participant in the study. 

Response: We have obtained participant’s signed consent and we have added the related statement in the Method section (see lines 118-119).

4. We noted in your submission details that a portion of your manuscript may have been presented or published elsewhere.

[A pilot of this study was presented at the Computer Supported Cooperative Work and Social Computing Conference in 2017 (please the Related Manuscript). Since that time we have collected additional data, including the addition of Condition 2 in which participants collaborated only via audio. (please see the table of the cover letter for overlapping variables).]

Response: Please see our detailed response in the cover letter. 

Reviewer #1: This is a thoughtful, elegant manuscript that meets (and exceeds) the bar for publication--and also offers really interesting, counterintuitive, and practical implications. It is technically sound, and the authors are clearly experts in this field. Excellent submission.

Response: Thank you for your kind words.

Reviewer #2: #Review of manuscript PONE-D-20-24495 for PLOS ONE

The authors of this work report the results of an experimental study in which they investigate the verbal and nonverbal behaviour and task performance of dyads who carry out several tasks while in different locations (“distributed collaboration”) and connected via audio-only or using concurrent auditory and visual modalities (through live video). They observe that the degree of prosodic synchrony within dyads in the audio-only condition was higher compared to the degree of prosodic synchrony in the audio+video group. No overall differences in collective intelligence measures and facial expression synchrony were observed between the two groups/conditions, but higher prosodic synchrony correlated with higher collective intelligence. The authors argue that the presence of video may thus not only have benefits (as suggested by previous work and anecdotal evidence) but also drawbacks for communication and collaboration, and highlight the societal relevance of these findings in the current digital era. This research is timely, the manuscript is well-written, and I agree that the findings have clear societal relevance. Several questions for clarification keep me from recommending acceptance of the manuscript in its current form.

Response: Thank you for your kind comments and also for your suggestions to improve our manuscript toward publication. 

Minor comments and suggestions

1. The word ‘disrupts’ in the title is quite strong in the absence of a causal effect of the presence of video and in light of the finding that “there was not a significant difference in the observed level of collective intelligence” across the two conditions/groups (line 224).

Response: This is an excellent point, and we thank you for raising it. In reconsidering the title based on your feedback we have decided to alter it to more accurately reflect the chain of relationships we observe in the data. This results in a slightly longer but more accurately descriptive title which we hope you will agree is more appropriate. 

The new title is: “Speaking out of turn: How video conferencing reduces vocal synchrony and collective intelligence”

2. The authors theoretically contrast their set-up with earlier studies that investigated similar issues in ‘face-to-face’ set-ups (see e.g. the abstract, and line 46-48). However one could argue that also in the authors’ video condition, participants are sitting face-to-face – they are just in different locations (see also our description in the laboratory published protocol). I would therefore suggest being more careful when using the term ‘face-to-face’ or avoiding it at all.

Response: Thank you for raising this point, which highlighted for us the need to clarify our procedure. In our study, participants were taken to physically separated rooms from the very beginning of the experiment (see also the subsection Participant Recruitment and Data Collection, lines 121- 134), and could only see and/or hear each other via the small screen-based video-conference window and audio available via their computer. In the literature, “face-to-face” interactions refer to physically collocated interactions not mediated in any way by technology (e.g., Crowley and Mitchell, 1994 ; Nardi & Whitaker, 2002) which allow collaborators to access much more nonverbal cues than simply seeing a fairly low-resolution video of their collaboration partner’s face. So we believe our study does appropriately operationalize a remote collaboration scenario distinct from face-to-face setups investigated in prior research.

Crowley, D. J., & Mitchell, D. (Eds.). (1994). Communication theory today. Stanford University Press.

Nardi, B. A., & Whittaker, S. (2002). The place of face-to-face communication in distributed work. Distributed work, 83, 112. 

3. The work by Levinson on turn-taking is relevant to the discussion of turn-taking in the current manuscript and I would suggest referring to it as it has been highly influential. See for instance:

Levinson, S. C. (2016). Turn-taking in human communication–origins and implications for language processing. Trends in Cognitive Sciences, 20(1), 6-14.

Response: Thank you for suggesting this important reference. We have now incorporated Levinson’s work into our discussion of turn-taking. Please see line 49. 

4. The authors’ focus on synchronization of facial expressions (line 79) comes a bit as a surprise to the reader at this point in the text. I would suggest adding a paragraph prior to the current line 79 in which earlier work on facial expression synchrony is discussed more explicitly.

Response: Thank you for your comment. In thinking more about this, we now introduce synchrony in facial expressions (and vocal cues) in lines 31-33. 

5. The authors explain that they used six tasks to measure collective intelligence. More information is needed here in the main text. Which six tasks were used exactly and what was the procedure used in each task? What number (mean unweighted score) corresponds to a high score on those tasks and what corresponds to a relatively low score?

Response: We used six tasks (typing, matrix solving, sudoku, unscramble words, memory, brainstorming) that represent executing, remembering, generating, and choosing, respectively. Participants were logged onto the Platform for Online Group Studies (POGS) with the username Participant A or Participant B and worked synchronously together on each of the tasks. The POGS system was used to automatically administer the tasks (e.g., presenting instructions and task interface, regulating time). We provide detailed information for the procedure in our procedure section as well as in our published protocol. In addition, we provide a table describing the nature of tasks, scoring rules, and descriptive statistics of task scores (see file in our protocol and below for your convenience). 

---see table in the original letter to the reviewers--

6. In the absence of more explicit information about the tasks that were used, I found it hard to understand why speaking turn inequality (see section 2.2.5) would be a theoretically relevant measure/proxy of a dyad’s spoken communication quality/performance. I would clarify more extensively early on in the manuscript that it has previously been related to collective intelligence, as is now explained in lines 251-252.

Response: We describe in greater detail now previous findings associating speaking equality and collective intelligence. Please see lines 44-51.

7. Very minor stylistic comments: the words co-located and collocated are used interchangeably throughout the manuscript. The final sentence of the abstract seems ungrammatical.

Response: We have now consistently used the word collocated. The verb “call” refers to the findings and not to the non-verbal synchrony so we believe the sentence is correct in the last sentence of our abstract. 

Reviewer #3: The authors investigate the effect of utilizing video versus non-video (audio only) communication on collective intelligence in dyads. Their investigation is nuanced, analyzing the dynamics of mediating and moderating factors in this process.

Response: Thank you.

The design of the experiment is clearly explained, and is sound. The same is true of the analytical strategy.

Response: Thank you.

The results of the study are counterintuitive, but are well-explained by the authors. The idea that the presence of video may actually hinder communication is a fascinating one, and the implications are important. As the authors mention in their discussion of future directions, it will be interesting to see whether these results generalize to larger groups.

Response: Thank you and we agree about future research in larger working groups. 

In the discussion section, I would like to see a little more discussion/elaboration on the finding displayed in Figure 3. If video (versus audio only) may increase the importance of social perceptiveness (due to its effect on facial expression synchrony), this is an important dynamic for people and organizations to take into consideration.

Response: Dear Reviewer, thank you for the kind feedback. Related to your specific comment here we have now elaborated on our findings related to Figure 3 and connected with some relevant research (please see lines 279-285). Thank you!

I enjoyed reading this work - best of luck to all of the authors in the future!

Response: Thank you!

---

## [Editor Report · Decision Letter 1]

11 Feb 2021

Speaking out of turn: How video conferencing reduces vocal synchrony and collective intelligence

PONE-D-20-24495R1

Dear Dr. Tomprou,

We’re pleased to inform you that your manuscript has been judged scientifically suitable for publication and will be formally accepted for publication once it meets all outstanding technical requirements.

Kind regards,

Marcus Perlman, Ph.D

Academic Editor

PLOS ONE

Additional Editor Comments (optional):

Thank you for your detailed revisions in response to the reviewers' comments. I am happy to accept the article for publication.
---

## [Editor Report · Acceptance letter]

4 Mar 2021

PONE-D-20-24495R1 

Speaking out of turn: How video conferencing reduces vocal synchrony and collective intelligence 

Dear Dr. Tomprou:

I'm pleased to inform you that your manuscript has been deemed suitable for publication in PLOS ONE. Congratulations! Your manuscript is now with our production department. 

Kind regards, 

on behalf of

Dr. Marcus Perlman 

Academic Editor

PLOS ONE